# Therapeutic Repurposing of Sertraline: Evidence for Its Antifungal Activity from In Vitro, In Vivo, and Clinical Studies

**DOI:** 10.3390/microorganisms13102334

**Published:** 2025-10-10

**Authors:** Carmen Rodríguez-Cerdeira, Westley Eckhardt

**Affiliations:** 1Department of University Program for Seniors, University of Vigo, E.E. Industrial Rúa Torrecedeira 86, 36201 Vigo, Spain; m72433337@gmail.com; 2Fundación Vithas, Vithas Foundation, 28043 Madrid, Spain; 3Dermatology Department, Hospital CMQ-Concheiro, Manuel Olivié 11, 36203 Vigo, Spain

**Keywords:** sertraline, antifungal activity, drug repurposing, *Candida* spp., *Cryptococcus neoformans*

## Abstract

Sertraline, a selective serotonin reuptake inhibitor (SSRI), has emerged as a candidate for therapeutic repurposing due to its reported antifungal activity. We systematically reviewed in vitro, in vivo, and clinical evidence up to July 2025 (PubMed, Scopus, Web of Science). As a result, 322 records were screened and 63 studies were found to meet the inclusion criteria (PRISMA 2020). We close a critical gap by consolidating relevant evidence on *Candida auris*, including preclinical in vivo models, which have been under-represented in previous summaries. Outcomes included minimum inhibitory and fungicidal concentrations (MIC/MFC), biofilm inhibition, fungal burden, survival, and pharmacokinetic/pharmacodynamic parameters. Preclinical data indicate its activity against clinically relevant fungi—particularly *Cryptococcus neoformans* and *Candida* spp., including *C. auris*—as well as consistent anti-biofilm effects and synergy with amphotericin B, fluconazole, micafungin, or voriconazole. Mechanistic evidence implicates mitochondrial dysfunction, membrane perturbation, impaired protein synthesis, and calcium homeostasis disruption. However, its potential for clinical translation remains uncertain: in cryptococcal meningitis, small phase II studies suggested improved early fungicidal activity, whereas a phase III randomized trial did not demonstrate a benefit regarding survival. Pharmacokinetic constraints at conventional doses, the absence of an intravenous formulation, and safety considerations at higher doses further limit its immediate applicability. Overall, the available evidence supports sertraline as a promising adjuvant candidate, rather than a stand-alone antifungal. Future research should define PK/PD targets, optimize doses and formulations, and evaluate rational combinations through rigorously designed trials, particularly for multidrug-resistant and biofilm-associated infections.

## 1. Introduction

Sertraline (SRT) is a selective serotonin reuptake inhibitor (SSRI) that is widely used as a first-line antidepressant for major depressive disorder and several anxiety spectrum conditions, supported by a favorable pharmacological profile and predictable kinetics [1,2,3]. Beyond neuromodulation, experimental works have reported its antifungal activities against clinically relevant pathogens—*Candida albicans*, *Cryptococcus neoformans*, *Aspergillus fumigatus*, *Trichophyton rubrum*, among others—prompting interest in drug repurposing due to resistance limiting the currently available options [4,5].

From a pharmacological standpoint, SRT blocks presynaptic 5-HT reuptake and is generally well tolerated; it has an elimination half-life of ~26 h and is hepatically metabolized predominantly via CYP2B6, CYP2C19, and CYP2D6 [6,7]. Screening studies performed in the early 2010s identified its activity against *C. neoformans* and later against *Candida* spp. and dermatophytes, broadening its potential antifungal spectrum [8,9]. Unlike azoles, echinocandins, or amphotericin B, SRT appears to act through multiple, non-exclusive mechanisms; for brevity and to avoid redundancy, these are summarized in Table 2, (e.g., mitochondrial dysfunction, membrane perturbation, impairment of protein synthesis, and calcium homeostasis disruption) with supporting citations [10,11,12,13,14].

Clinically, evaluations of SRT have focused on cryptococcal meningitis in combination regimens (often with fluconazole). While initial evidence suggested improved early fungicidal activity, a phase III randomized trial did not demonstrate a benefit regarding survival, underscoring translational constraints [15]. SRT’s central nervous system (CNS) penetration ability has led to sustained interest with respect to CNS mycoses and, as such, it may provide certain benefits in settings with constrained access to amphotericin B. However, its proposed adjuvant roles remain hypothesis-generating and require confirmation through rigorously designed studies [16].

The pharmacological profile of SRT, including its pharmacodynamics, metabolism, and pharmacokinetics, is summarized in Table 1.

**Table 1 microorganisms-13-02334-t001:** Pharmacological properties of sertraline.

Focus	Key Findings	Ref.
Mechanisms of antifungal resistance	Target mutations (ERG11/FKS), efflux (ABC/MFS), biofilm tolerance, Hsp90–calcineurin stress response, genomic plasticity/aneuploidy.	[4]
Pharmacodynamics	SSRI; increases synaptic serotonin via reuptake inhibition.	[6]
Pharmacokinetics	Metabolized by CYP2B6/CYP2C19/CYP2D6; steady state ~7 days.	[7]
Metabolism	N-desmethylsertraline; half-life 24–26 h; once-daily dosing.	[17]
Exposure vs. MIC	Conventional doses may not reach antifungal MICs.	[18]
Repurposing	Antifungal potential noted; activity vs. *Cryptococcus neoformans*.	[8]

ERG11: Lanosterol 14-α-demethylase gene (azole target); FKS1/2: β-1,3-glucan synthase catalytic subunits (echinocandin target); ABC: ATP-binding cassette efflux transporters; MFS: Major facilitator superfamily efflux transporter; biofilm tolerance: reduced drug susceptibility of biofilm-embedded cells (non-heritable); Hsp90–calcineurin stress response: Chaperone–phosphatase axis supporting tolerance; genomic plasticity/aneuploidy: adaptive karyotypic changes (chromosome copy number variation); SSRI: Selective serotonin reuptake inhibitor; CYP2B6/CYP2C19/CYP2D6: Cytochrome P450 isoenzymes; MIC: Minimum inhibitory concentration.

## 2. Materials and Methods

### 2.1. Study Design and Reporting Framework

This systematic review was conducted and reported in accordance with the PRISMA 2020 guidelines [19]. The protocol was not registered (e.g., PROSPERO).

### 2.2. Information Sources and Search Strategy

We searched PubMed, Scopus, and Web of Science from database inception to 31 July 2025. Searches combined controlled vocabulary and free-text terms related to SRT and fungal infections. An example Boolean string was as follows: (SER OR “N-desmethylSRT” OR SRT) AND (fung* OR Candida OR “Candida auris” OR “Cryptococcus neoformans” OR cryptococc* OR aspergill* OR biofilm OR antifungal OR “drug repurposing”). Reference lists of included studies and relevant reviews were hand-searched to identify additional records. No automation tools were used for searching or screening.

### 2.3. Eligibility Criteria

For inclusion, we considered in vitro, in vivo (preclinical), and clinical studies evaluating SRT alone or in combination with licensed antifungals and reporting ≥ 1 antifungal outcome: minimum inhibitory concentration/minimum fungicidal concentration (MIC/MFC), time–kill dynamics, biofilm metrics (biomass/viability/formation), fungal burden reduction, survival, or PK/PD parameters. Articles in English or Spanish were considered to be eligible. For exclusion, editorials, narrative reviews, conference abstracts without extractable data, non-antifungal outcomes, wrong organism/model/population, duplicate datasets/overlap, or insufficient/irretrievable data were screened.

### 2.4. Study Selection

Records were exported to EndNote 20, and duplicates were removed using software-based algorithms followed by manual verification. Two reviewers independently screened titles/abstracts, then full texts; disagreements were resolved by consensus. After de-duplication, 322 records were screened, 63 of which met the inclusion criteria. Reasons for full-text exclusion (e.g., non-antifungal outcome, wrong organism/model, duplicate dataset, insufficient data) are summarized in the PRISMA 2020 flow diagram (Figure 1) [19].

### 2.5. Data Extraction

Two reviewers independently extracted data using a standardized form: study type, organism(s), assay/model, SRT dose/concentration, partner antifungals (if any), endpoints (MIC/MFC, time–kill, FIC index or ε for synergy, biofilm metrics, fungal burden, survival, PK/PD), and clinical safety/tolerability, where available. Disagreements were resolved by consensus.

### 2.6. Outcomes and Operational Definitions

Synergy (checkerboard): Fractional inhibitory concentration (FIC) index ≤ 0.5; antagonism: FIC ≥ 4.Synergy (time–kill): ≥2 log10 CFU/mL reduction in the combination vs. the most active single agent at matched time points.Biofilm outcomes: Changes in biomass or viability (e.g., crystal violet/XTT), or inhibition of biofilm formation vs. appropriate controls.

These definitions were applied as reported or recalculated when sufficient data were provided.

### 2.7. Risk of Bias and Certainty

Given methodological heterogeneity, we performed a qualitative appraisal using RoB 2 for randomized trials, ROBINS-I for non-randomized clinical studies, and domain-based considerations for preclinical work (randomization/blinding, sample size, outcome reporting). No meta-analysis was performed due to clinical and methodological heterogeneity.

### 2.8. Synthesis Methods

The findings were synthesized narratively and organized by evidence tier. Summary tables present pharmacological properties (Table 1), proposed mechanisms (Table 2), preclinical activities (Table 3), clinical evidence (Table 4), pharmacodynamic interactions (Table 5), and safety/barriers/future directions (Table 6). The counts at each screening stage and reasons for exclusion are detailed in Figure 1 (PRISMA 2020 flow diagram) [19].

### 2.9. Ethics

This review is based on previously published data and did not involve new studies with humans or animals; therefore, ethical approval was not required.

**Figure 1 microorganisms-13-02334-f001:**
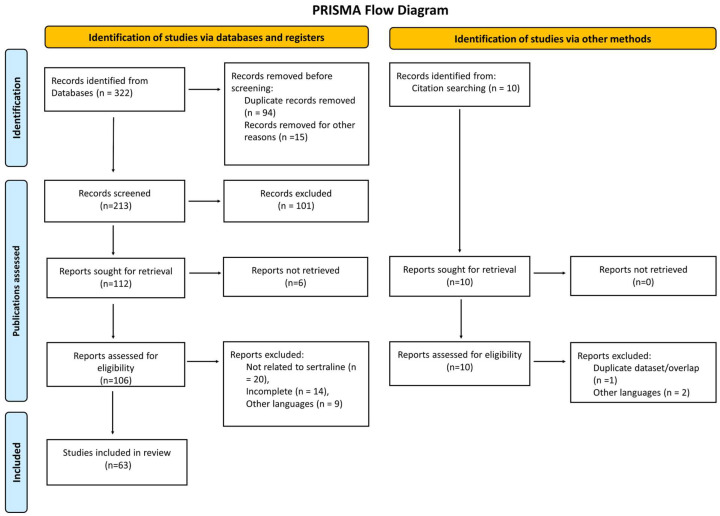
PRISMA flow diagram for the systematic review of the antifungal properties of SRT, including database and register searches. The number of records from each source is specified. No automation tools were used in the screening process.

## 3. Results

The reviewed studies consistently demonstrated that SRT, a selective serotonin reuptake inhibitor (SSRI), exhibits antifungal activity in vitro and in vivo against clinically important pathogens. To facilitate readability and avoid redundancy, detailed outcomes are summarized in Table 3 and Table 4; pharmacodynamic interactions are listed in Table 5; and the proposed mechanisms are summarized separately in Table 2. The results are grouped into four areas: activity against *Cryptococcus neoformans* [20], effects on *Candida* spp. [21], intracranial pharmacokinetics [22], and population-based observational data [23]. The proposed antifungal mechanisms of action are summarized in Table 2.

In an open-label, non-randomized phase II clinical trial in Tanzania [24], SRT (400 mg/day) was evaluated as an adjunctive therapy for cryptococcal meningitis treated with fluconazole (1200 mg/day), with or without a short course of amphotericin B (0.7–1 mg/kg for 5 days). Among 46 patients, two-week survival was 64% without amphotericin and 89% with amphotericin; at ten weeks, survival was 21% and 61%, respectively (*p* = 0.012). Early fungicidal activity (EFA) in CSF improved with amphotericin (0.264 vs. 0.473 log10 CFU/mL/day; *p* = 0.03) [24,25]. The main therapeutic applications and clinical outcomes are presented in Table 3.

Preclinical murine models of *C. neoformans* infection showed that SRT (15 mg/kg/day) reduced fungal burden in the brain and spleen, with results comparable with fluconazole; meanwhile, the SRT + fluconazole combination further reduced kidney burden and improved survival [26]. In vitro, 153 clinical isolates of *Cryptococcus* were found to be susceptible to SRT, with MIC90 < 10 µg/mL [27]. These in vitro and in vivo findings are compiled in Table 4.

SRT inhibited mature and developing biofilms of *Candida albicans*, *C. tropicalis*, *C. parapsilosis*, *C. glabrata*, and *C. auris* [28]. In a population-based cohort of >77,000 women, reported vaginal candidiasis was lower with antidepressants (including SRT) than with antibiotics (0.15% vs. 0.6%) [23]. This finding was further supported by additional antifungal and antibiofilm studies on *Candida* species [29]. In *C. auris*, SRT inhibited biofilm formation by up to 71% and altered morphogenesis and membrane integrity, supporting its evaluation as an adjuvant against azole-resistant strains, especially in persistent or device-associated infections [30,31]. Reductions approaching ~80% in cell viability within established *Candida* biofilms were observed at sub-cytotoxic concentrations, and SRT also prevented biofilm formation in a dose- and time-dependent manner (6–48 h) [30]. The synergistic and antagonistic interactions of sertraline with licensed antifungals are summarized in Table 5.

In a multicenter study conducted in northeastern Mexico testing 29 clinical isolates of non-*Aspergillus* multidrug-resistant molds (including *Lomentospora prolificans*, *Scedosporium* spp., *Fusarium* spp., *Alternaria* spp.), SRT showed in vitro activity and synergy with amphotericin B against several taxa, whereas its antagonism with voriconazole was noted for *Purpureocillium lilacinum*, underscoring the need for careful combination selection [32]. Subsequent studies expanded these observations and confirmed the broader antifungal potential of SRT under similar experimental conditions [33,34]. Clinical signals remain heterogeneous, reinforcing the rationale for adjunctive strategies guided by PK/PD-informed dosing and targeted drug combinations, as summarized in the tables above.

**Table 2 microorganisms-13-02334-t002:** Proposed antifungal mechanisms of action of sertraline.

Mechanism of Action	Target Organism(s)/Context	Key Notes	Ref.
β-1,3-glucan synthesis blockade	*Candida* spp. (cell wall)	Cell wall impairment	[10]
Membrane lipid interaction/disruption	*Candida* spp.	Loss of integrity, lipid bilayer destabilization	[12,14]
Oxidative stress induction (↑ ROS)	*Candida* spp., *Cryptococcus neoformans*	ROS accumulation	[35]
Mitochondrial dysfunction	*Candida* spp., *Cryptococcus* spp.	↓ ATP production, apoptosis induction	[35,36]
Efflux pump downregulation (Cdr1p)	C. albicans	↑ Intracellular azole concentration	[37]
Ribosomal inhibition/↓ translation	*S. cerevisiae*, *Cryptococcus* spp.	Impaired protein synthesis, ↓ ribosome assembly	[13,38]
Calcium homeostasis disruption	Fungal cells	Abnormal Ca^2+^ influx, signaling alteration	[39,40]

ROS = Reactive oxygen species; ATP: Adenosine triphosphate. ↑ indicates an increase; ↓ indicates a decrease.

**Table 3 microorganisms-13-02334-t003:** Therapeutic applications of and clinical evidence for sertraline.

Study Type/Setting	Regimen (Alone/Combination)	Main Outcomes/Notes	Ref.
Clinical trial (Tanzania)	AMB 0.7–1 mg/kg × 5 days + SRT 400 mg/day + FLC 1200 mg/day	↑ Survival; ↑ CSF clearance in cryptococcal meningitis.	[24]
Clinical trial	Short-course AMB + SRT 400 mg/day + FLC 1200 mg/day	Improved outcomes in cryptococcal meningitis.	[25]
RCT (ASTRO-CM)	Adjunctive sertraline	No survival benefit; dosing/levels issues raised.	[41]
Rationale/PK	—	CNS penetration supports adjuvant role where AMB is limited.	[16]

AMB: Amphotericin B; SRT: Sertraline; FLC: Fluconazole; CSF: Cerebrospinal fluid. CNS: Central nervous system; RCT: Randomized controlled trial; ASTRO-CM: Adjunctive sertraline for the treatment of cryptococcal meningitis. ↑ indicates an increase.

**Table 4 microorganisms-13-02334-t004:** In vitro and in vivo antifungal activity of sertraline (preclinical).

Model	Pathogen(s)	Antifungal Partner(s)	Key Findings	Ref.
in vitro (biofilm)	*Candida* spp.	—	↓ viability > 80%; ALS3-mediated prevention.	[30]
in vitro (biofilm)	*C. auris*	—	~71% biofilm inhibition; morphogenesis/membrane effects.	[31]
in vitro	*C. auris*	FLU/MCF	Inhibition at clinically relevant levels; synergy.	[42]
in vivo (murine model)	*C. auris*	VOR (±)	Reduced fungal burden in vivo.	[43]
in vitro	*Cryptococcus neoformans*	—	High susceptibility to SRT.	[27]

ALS3: Agglutinin-like sequence 3; FLU = Fluconazole; MCF = Micafungin; VOR = Voriconazole; SRT = Sertraline. ↓ indicates a decrease.

**Table 5 microorganisms-13-02334-t005:** Synergistic or antagonistic interactions of sertraline with antifungals.

Combination (with SRT)	Pathogen(s)	Interaction	Ref.
FLU	*C. auris*	Synergy	[42]
FLU	*C. neoformans*	Adjunct regimen (no formal synergy metric)	[24]
VOR	*C. auris*	Synergy	[43]
MCF	*C. auris*	Synergy	[42]
AMB	*C. neoformans*	Synergy	[27]
VOR	*Purpureocillium lilacinum*	Antagonism	[32]

FLU = Fluconazole; VOR = Voriconazole; MCF = Micafungin; AMB = Amphotericin B.

Pharmacokinetic studies have indicated CNS accumulation of SRT approximately 20–40× that in plasma, a property relevant to cryptococcal meningitis and other CNS mycoses; however, whether sustained antifungal exposure can be achieved at conventional psychiatric doses remains uncertain [44]. Overall, the evidence supports SRT’s antifungal and antibiofilm activities, often synergizing with standard agents, while clinical signals remain heterogeneous, reinforcing the rationale for adjunctive strategies guided by PK/PD-informed dosing and targeted combinations (see Table 3, Table 4, Table 5 and Table 6; mechanisms in Table 2).

## 4. Discussion

Escalating antifungal resistance and the growing burden of invasive mycoses have renewed interest in drug repurposing as a pragmatic strategy to expand antifungal options with known safety profiles. Within this framework, SRT has attracted attention not only due to its CNS effects but also its antifungal properties documented across in vitro, in vivo, and preliminary clinical studies [45,46]. Such evidence spans multiple pathogens—*Cryptococcus neoformans*, *Candida albicans*, *Candida auris*, *Candida glabrata*, *Aspergillus fumigatus*, and *Fusarium* spp.—signaling a broad range of activity [47]. The MICs in the 1–8 µg/mL range reported for *C. neoformans* are noteworthy, given the enduring challenge of CNS cryptococcosis in immunocompromised hosts [27] (see Table 4 for models and Table 5 for pharmacodynamic interactions).

Preclinical models have consistently shown the CNS burden reduction and survival advantages associated with SRT, particularly when used in combination with standard agents, which is an effect plausibly aided by its ability to penetrate the CNS [45]. Outside classical mycoses, clinical observations are limited but hypothesis-generating: in premenstrual dysphoric disorder patients with recurrent Vulvovaginal candidiasis (VVC), symptom-free intervals during SRT treatment and relapse after withdrawal have been described [48]. Rapid, dose-dependent killing of *Candida* spp. has been reported in vitro (MFC 3–29 mg/mL; ≥99% kill at 2× MFC within 30 min) [49], and broader epidemiological studies have emphasized its clinical relevance to VVC and non-*albicans* species [42]. Regarding *C. auris,* an emerging multidrug-resistant pathogen, SRT inhibits its growth/biofilm formation and has demonstrated synergy with azoles or echinocandins, suggesting a possible adjuvant role in critical settings that warrants careful evaluation [43]. In a murine *C. auris* candidemia model, SRT produced dose-dependent kidney burden reductions and SRT + voriconazole outperformed either monotherapy as the first preclinical demonstration of this combination in *C. auris* [43,49].

Mechanistically, SRT does not recapitulate the targets of azoles, echinocandins, or amphotericin B. Converging data indicate a multifactorial profile involving mitochondrial dysfunction and oxidative stress; impairment of protein synthesis via ribosomal effects; membrane perturbation with potential efflux modulation; and altered calcium homeostasis [35,36,37,38,39,40] (Table 2). These non-exclusive mechanisms may act complementarily, offering a plausible basis for the synergy observed with standard antifungals and a potentially lower risk of emergent resistance [46]. For detail and to avoid redundancy, the mechanisms are summarized in Table 2.

Balanced consideration of the available clinical evidence is essential. Phase II studies (Uganda/Tanzania) have suggested improved early fungicidal activity when SRT was added to standard therapy in cryptococcal meningitis [50,51], which motivated the adjunctive use of sertraline for the treatment of cryptococcal meningitis (ASTRO-CM) randomized trial [52]. However, the ASTRO-CM trial did not demonstrate a 10-week benefit regarding survival, highlighting a translational gap between microbiological evidence and clinical outcomes [41]. Plausible explanations include suboptimal dosing, adherence issues, inter-individual PK variability, and the difficulty of sustaining effective CSF exposure at psychiatric doses [41]. Accordingly, SRT should be viewed as a candidate adjuvant and not as a first-line antifungal monotherapy (see Table 3 for clinical evidence and regimens).

Pharmacological properties help to explain this profile. Its high lipophilicity, ~44% oral bioavailability, extensive protein binding, and large volume of distribution favor tissue (including CNS) penetration but complicate prediction of free active drug levels at infection sites [53]. Hepatic metabolism via CYP2B6, CYP2C19, and CYP3A4 yields N-desmethylsertraline with minimal antifungal activity, and its 24–26 h half-life enables once-daily dosing [17]. At conventional doses (50–200 mg/day), plasma levels frequently remain below sustained antifungal MIC targets; therefore, doses of 300–400 mg/day with close monitoring or PK strategies have been proposed to increase exposure [15,18]. While its psychiatric safety profile is well characterized (GI effects, insomnia, headache, sexual dysfunction, rare serotonin syndrome, QTc prolongation, hyponatremia, withdrawal) [54], heightened vigilance is required for immunocompromised polymedicated patients. Interactions with CYP3A4-metabolized azoles (e.g., fluconazole, voriconazole) may increase its toxicity; paradoxically, such interactions could be leveraged to increase the exposure to SRT, but only under stringent clinical and pharmacological oversight [55] (see Table 1 for pharmacology and Table 6 for safety/barriers).

Against the backdrop of antifungal resistance, *C. auris* remains a global priority. The immediate availability and low cost of SRT are practical advantages, and reports of fluconazole-resistant strains retaining susceptibility suggest independent mechanisms and the possibility of re-sensitization via combined therapies [56,57]. Nonetheless, before its routine clinical use is considered, key limitations must be addressed: achieving therapeutic concentrations in systemic/deep-seated infections [15], the absence of an intravenous formulation for critical care settings [58], the lack of positive phase III evidence [15], and uncertain safety in non-psychiatric patients with fungal sepsis, neutropenia, or hepatic impairment [15]. Taken together, these considerations support pursuing SRT primarily as an adjunctive option in defined contexts and within the scope of rigorous protocols (see Table 5 for combinations and interaction patterns).

Priority directions for future research include (i) advanced PD studies to delineate concentration–effect relationships [59]; (ii) controlled trials focused on *C. auris*, *Aspergillus* spp., or mixed infections in immunosuppressed hosts [60]; (iii) synthesis of SRT analogs with greater antifungal specificity and reduced serotonergic activity [61]; (iv) the development of parenteral or liposomal formulations for treatment of invasive disease [62]; and (v) deeper investigation of biofilms, given their clinical relevance in catheters and implantable devices [63]. Overall, the available evidence supports the consideration of SRT as a promising adjunctive candidate, while its ultimate clinical utility hinges on closing the translational gap through clear PK/PD targets, rational combinations, and robust study designs (priority items summarized in Table 6).

**Table 6 microorganisms-13-02334-t006:** Safety profile, barriers, and future research directions regarding sertraline.

Category	Key Note/Recommendation	Ref.
Safety/tolerability	Generally well tolerated; rare risks: serotonin syndrome, QTc prolongation, hyponatremia.	[54]
Safety/interactions	Monitor adverse effects and CYP-mediated interactions in severe infections.	[55]
Barrier	Lack of IV formulation limits use in critically ill or rapidly progressive disease.	[58]
Future direction	Define concentration–response (PK/PD) for antifungal activity.	[59]
Future direction	Biofilm-focused and mixed-community studies.	[63]

QTc = Corrected QT Interval; CYP: Cytochrome P450; PK/PD: Pharmacokinetics/pharmacodynamics.

## 5. Conclusions

SRT demonstrates consistent antifungal and antibiofilm activity across diverse pathogens, with reproducible synergy when combined with standard antifungal agents. Preclinical and early clinical studies highlight its potential as an adjunctive therapy, particularly in cryptococcal meningitis and biofilm-associated infections, while also revealing major translational barriers. Pharmacokinetic limitations at conventional doses, the lack of intravenous formulations, and safety concerns in critically ill or polymedicated patients currently restrict its use. Taken together, these findings support positioning SRT as a promising adjunctive candidate rather than a first-line antifungal. Future research should prioritize optimized dosing strategies, improved formulations, and rigorously designed clinical trials to determine its true therapeutic role within the antifungal armamentarium.

## 6. Future Perspectives (Considering Current Limitations)

Given the exposure shortfalls at conventional doses, the lack of an intravenous formulation, heterogeneous clinical signals, and potential drug–drug interactions in polymedicated immunocompromised patients, future work should (i) define PK/PD targets and exposure thresholds in plasma and CSF, incorporating therapeutic drug monitoring approaches; (ii) evaluate optimized doses and safety at higher exposures; (iii) develop parenteral or targeted delivery formulations; (iv) conduct randomized, adequately powered trials focused on defined use cases such as *Candida auris* bloodstream or device-associated infections and cryptococcal meningitis, including rational combination arms informed by mechanistic and pharmacodynamic insights; (v) deepen biofilm- and device-related models; (vi) characterize the resistance potential under SRT pressure; and (vii) implement pharmacovigilance measures in high-risk populations.

## Data Availability

No new data were created or analyzed in this study. Data sharing is not applicable to this article.

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
