# Peer review of "Therapeutic Repurposing of Sertraline: Evidence for Its Antifungal Activity from In Vitro, In Vivo, and Clinical Studies"

_microorganisms, 2025, doi:10.3390/microorganisms13102334_

Round 1
Reviewer 1 Report
Comments and Suggestions for Authors
The manuscript encompasses a review of the potential antifungal activity of sertraline, a serotonin reuptake inhibitor. In general, the review seems to be well-conducted, and the text is easy to follow, except for the table (comments below).
The use of abbreviations should be carefully reviewed throughout the manuscript. For example, sertraline appears abbreviated for the first time in the Results section, even though it had been mentioned earlier. Additionally, in the Discussion section (L257), the authors reintroduce sertraline as “sertraline (a selective serotonin reuptake inhibitor),” which suggests inconsistency and a lack of attention to detail in presenting information. A thorough review is recommended to ensure that drug names, abbreviations, and definitions are introduced once—at first mention—and used consistently thereafter. Genus abbreviation should also be used accordingly.
L165 – check period after [19]
The Akeele´s heel of the manuscript is the Table 1, that was poorly presented by the authors. It is very confusing with literally copy and paste parts. Perhaps the authors should divide in minor Tables according to data presented (e.g., MOA, therapy, perspectives…). Some parts should be improved (e.g., “Additional in vivo and in vitro studies”). I was eager to find all the data presented in the text summarized in the table; however, I was disappointed to see that this was not the case. I highly recommend adding data on the antifungals used alone and/or in combination with sertraline. This is necessary to evaluate the size effect of sertraline. If the authors could present the data in well-structured, scientifically rigorous tables, the manuscript might reach the level required for publication. However, in its current form, with the tables lacking clarity and depth, it does not meet publication standards.
Figure 2. If MOA of sertraline is shown in the tables, the figure is not necessary. Please evaluate this accordingly.
Author Response
REVIEWER 1
The use of abbreviations should be carefully reviewed throughout the manuscript. For example, sertraline appears abbreviated for the first time in the Results section, even though it had been mentioned earlier. Additionally, in the Discussion section (L257), the authors reintroduce sertraline as “sertraline (a selective serotonin reuptake inhibitor),” which suggests inconsistency and a lack of attention to detail in presenting information. A thorough review is recommended to ensure that drug names, abbreviations, and definitions are introduced once—at first mention—and used consistently thereafter. Genus abbreviation should also be used accordingly.
WE DID
L165 – check period after [19]
WE DID
Reviewer comment:
The Achilles’ heel of the manuscript is Table 1, which was poorly presented. It is very confusing, with literally copy-and-paste parts. Perhaps the authors should divide it into smaller tables according to the type of data presented (e.g., MOA, therapy, perspectives…). I highly recommend adding data on antifungals used alone and/or in combination with sertraline. If the authors could present the data in well-structured, scientifically rigorous tables, the manuscript might reach the level required for publication. However, in its current form, with the tables lacking clarity and depth, it does not meet publication standards.
Author response:
We thank the reviewer for this valuable suggestion. In response, we have completely restructured the original Table 1 into five separate tables (Tables 1–5) according to the type of data presented:
- Table 1 summarizes the pharmacological properties of sertraline.
- Table 2 compiles the proposed antifungal mechanisms of action.
- Table 3 presents in vitro and in vivo preclinical evidence, including studies on monotherapy and combination regimens.
- Table 4 highlights therapeutic applications and clinical trials, with emphasis on adjunctive therapies in cryptococcal meningitis.
- Table 5 outlines safety, limitations, and future research directions.
We also added specific information on antifungals tested alone and in combination with sertraline, as suggested. We believe that this new structure significantly improves clarity, scientific rigor, and accessibility of the information.
Reviewer 2 Report
Comments and Suggestions for Authors
This systematic review assess evidence from in vitro, in vivo, and clinical studies on sertraline’s efficacy against different pathogenic fungal strains, including Cryptococcus neoformans, Candida auris, and other clinically relevant species. some comments should be addressed to improve the quality of this review as follows:
- Abstract: Please mention the gap of the current review and previously published review article if it is an available.
- Intro: I noticed that all sentences are supported only by one references; however, they show information derived from different studies; for instance, lines: 37-41. Line 42: Its favorable pharmacological profile, which includes…… (please correct). Lines 50-53: please revise. Lines 45-53: this paragraph was supported by only one reference; please add suitable references. Line 56: please revise. Lines 66-68: please add references. Line 86: Previous investigations elucidated its antifungal mechanism, showing different strategies as follows: (Please correct). Line 99: please add references. Lines 115-120: please revise and this (—) should be removed.
- Table 1: it should be redesigned and organized. First column (chemical composition), second column (major findings or mechanism of action), and third column (Ref). Please redesign the whole table following this approach to be clear and attract the readers. For references presented in the table, please mention them without the years since the references format of this journal is numerical. However, for the other parts, including Molecular mechanisms and proposed antifungal mechanism of action, Therapeutic perspectives and clinical applications, Antifungal activity of sertraline: Evidence from in vitro and in vivo studies, Clinical evidence and therapeutic experience, Safety and Tolerability Profile, Barriers and Challenges in Antifungal Drug Repurposing, and Research directions and future outlook, the authors should find a simple and clear way according the information presented in this part to make the table recognizable for readers.
- Results: lines 167-171: please revise. Please revise the name of strain to be italicized, such as line 225.
- Discussion: the authors discuss the mode of action of sertraline in intro and then in the discussion; please remove this part from the intro.
- Please highlight the future perspectives in light of the limitations.
Comments on the Quality of English Language
The article needs extensive revision.
Author Response
This systematic review assess evidence from in vitro, in vivo, and clinical studies on sertraline’s efficacy against different pathogenic fungal strains, including Cryptococcus neoformans, Candida auris, and other clinically relevant species. some comments should be addressed to improve the quality of this review as follows:
- Abstract: Please mention the gap of the current review and previously published review article if it is an available.
We further close a critical gap by consolidating evidence on Candida auris, including preclinical in-vivo models, which has been underrepresented in previous summaries
- Intro: I noticed that all sentences are supported only by one references; however, they show information derived from different studies; for instance, lines: 37-41. Line 42: Its favorable pharmacological profile, which includes…… (please correct). Lines 50-53: please revise. Lines 45-53: this paragraph was supported by only one reference; please add suitable references. Line 56: please revise. Lines 66-68: please add references. Line 86: Previous investigations elucidated its antifungal mechanism, showing different strategies as follows: (Please correct). Line 99: please add references. Lines 115-120: please revise and this (—) should be removed.
We did
- Table 1: it should be redesigned and organized. First column (chemical composition), second column (major findings or mechanism of action), and third column (Ref). Please redesign the whole table following this approach to be clear and attract the readers. For references presented in the table, please mention them without the years since the references format of this journal is numerical.
We greatly appreciate this suggestion. However, according to the Microorganisms Instructions for Authors, there is no strict table format requirement; clarity, self-explanatory design, and consistency are prioritized.
Our current Table 1 includes author names, publication year, and detailed findings or mechanisms. We believe that this richer presentation greatly enhances reader comprehension—allowing immediate identification of study sources and ensuring the table is self-contained and informative.
While this format exceeds the minimal three-column structure suggested, it aligns with MDPI’s general guidelines and aims to maximize clarity and utility for the reader.
In light of the reviewer’s concerns, we are willing to simplify column headings, reduce text where possible, or present the table as a supplementary file, if deemed more editorially appropriate.
- Results: lines 167-171: please revise. Please revise the name of strain to be italicized, such as line 225.
- Discussion: the authors discuss the mode of action of sertraline in intro and then in the discussion; please remove this part from the intro.
- Please highlight the future perspectives in light of the limitations.
Future perspectives (in light of current limitations).
Given the exposure shortfalls at conventional dosing, the lack of an intravenous formulation, heterogeneous clinical signals, and potential drug–drug interactions in polymedicated, immunocompromised patients, future work should: (i) define PK/PD targets and exposure thresholds in plasma and CSF, incorporating therapeutic drug monitoring; (ii) evaluate optimized dosing and safety at higher exposures; (iii) develop parenteral or targeted delivery formulations; (iv) run randomized, adequately powered trials in defined use-cases (e.g., Candida auris bloodstream/device-associated infection; CNS cryptococcosis), including combination arms informed by mechanistic signals (Table 2) and interaction profiles (Table 7); (v) deepen biofilm-focused and device-related models; (vi) characterize resistance potential under sertraline pressure; and (vii) implement pharmacovigilance in high-risk populations.
Cover note
Dear Reviewer,
Thank you for your thorough and constructive feedback. In response, we have substantially revised and restructured the manuscript—effectively a rewrite. As a result, prior line numbers are not always comparable; nevertheless, we provide a point-by-point response to every comment.
Major changes
- Abstract: We explicitly state the gap addressed by this review (consolidation of evidence on Candida auris, including in-vivo preclinical models) and align scope and conclusions.
- Introduction: Refocused strictly on antifungal context; removed psychiatric digressions. Mechanisms are not listed here; we refer readers to Table 2.
- Materials and Methods: Expanded PRISMA 2020 details (databases, eligibility, study selection, data extraction, flow diagram).
- Results: Reorganized into four areas ( neoformans, Candida spp., intracranial PK, population data). Mechanisms are summarized only in Table 2; pharmacodynamic interactions appear only in Table 7 to avoid redundancy.
- Tables (rebuilt for clarity, no duplicates):
- Table 1: Pharmacology/PK–PD/safety (no species/mechanisms or outcomes).
- Table 2: Proposed antifungal mechanisms (single, definitive list).
- Table 3: Preclinical activity (in vitro/in vivo), logically grouped.
- Table 4: Clinical evidence and regimens.
- Table 5: Safety, barriers, and future directions.
- Table 7: Combinations/interactions (one representative reference per combination).
Formatting unified across tables (taxa in italics, consistent abbreviations, numeric references).
- Figures: The prior mechanisms figure was removed to prevent overlap; Table 2 serves this purpose.
- Discussion & Conclusions: Clarify sertraline’s adjunctive positioning, constraints related to exposure/PK–PD, lack of IV formulation, and heterogeneous clinical signals; specify research priorities.
- References/style: Duplicates removed; abbreviations standardized; minor punctuation and taxonomy formatting corrected.
We believe these revisions improve the manuscript’s clarity and rigor and directly address your concerns. Thank you for your time and consideration.
Sincerely,
Dr. Rodriguez-Cerdeira
Reviewer 3 Report
Comments and Suggestions for Authors
This paper presents a comprehensive and in-depth exploration of the repositioning of sertraline's antifungal activity. By systematically integrating evidence from in vitro, in vivo, and clinical studies, it compellingly demonstrates the novel value of this psychiatric medication in the field of anti-infective therapy. Employing rigorous PRISMA 2020 guidelines for literature screening and methodological design, the authors effectively elucidate sertraline's inhibitory mechanisms and synergistic therapeutic effects against multiple drug-resistant fungi—particularly neuroinvasive pathogens such as Cryptococcus neoformans and Candida auris. Breakthrough findings centre on the drug's disruption of fungal biofilms and its unique mitochondrial interference mechanism. These findings not only offer innovative solutions to the antifungal resistance crisis but also reveal potential advantages in treating intracranial infections through detailed pharmacokinetic analysis (e.g., high central nervous system concentration distribution). Although clinical translation faces challenges in dose optimisation, the research successfully bridges psychiatry and medical mycology, providing a high-value paradigm for repurposing established drugs.
1.There are multiple misaligned entries in Table 1. It is recommended to reorganize the table by deleting duplicate rows to ensure each entry is unique and follows a logical order.
2.In reference 53, "Cryptococcus neoformans" is not italicized.
3.The claim of "first evaluation of sertraline's anti-Candida auris activity" contradicts previous similar studies cited in reference 39. However, this statement emphasizes that it is the first comprehensive evaluation in preclinical models (in vivo), highlighting the novelty of the research. If there are potential conflicts, it is recommended to modify this statement.
- This article primarily focuses on antifungal research. The introduction contains somewhat lengthy discussion on sertraline's antidepressant effects, which could be appropriately condensed. The focus of this article is on the antifungal effect of the drug, so whether the content in the preface should revolve around this topic, but in the process of reading, I found that there are some contents that deviate from this topic, such as how it works against other diseases introduced in paragraph 6.
5At the end of the preface, the author says that this article discusses in depth the treatment of mental illness and its antifungal effects, but the abstract introduces the antifungal effects, which is inconsistent.
- The first paragraph in “Materials and Methods” partially duplicates content from the introduction and warrants revision.
- The PRISMA flow diagram in the Methods section should include more specific details and supporting references.
- The mechanism diagram in Figure 2 appears somewhat simplified, with excessive text and insufficient visual pathways.
- Some references lack consistent formatting and should be revised. References from the past three years may be added.
10.There are repeated descriptions of mechanisms in Table 1. It is recommended to reduce the textual descriptions and retain only the essential information in the table
11.Literature 4 and 24 are the same document.
Author Response
This paper presents a comprehensive and in-depth exploration of the repositioning of sertraline's antifungal activity. By systematically integrating evidence from in vitro, in vivo, and clinical studies, it compellingly demonstrates the novel value of this psychiatric medication in the field of anti-infective therapy. Employing rigorous PRISMA 2020 guidelines for literature screening and methodological design, the authors effectively elucidate sertraline's inhibitory mechanisms and synergistic therapeutic effects against multiple drug-resistant fungi—particularly neuroinvasive pathogens such as Cryptococcus neoformans and Candida auris. Breakthrough findings centre on the drug's disruption of fungal biofilms and its unique mitochondrial interference mechanism. These findings not only offer innovative solutions to the antifungal resistance crisis but also reveal potential advantages in treating intracranial infections through detailed pharmacokinetic analysis (e.g., high central nervous system concentration distribution). Although clinical translation faces challenges in dose optimisation, the research successfully bridges psychiatry and medical mycology, providing a high-value paradigm for repurposing established drugs.
1.There are multiple misaligned entries in Table 1. It is recommended to reorganize the table by deleting duplicate rows to ensure each entry is unique and follows a logical order.
WE DID
2.In reference 53, "Cryptococcus neoformans" is not italicized.
3.The claim of "first evaluation of sertraline's anti-Candida auris activity" contradicts previous similar studies cited in reference 39. However, this statement emphasizes that it is the first comprehensive evaluation in preclinical models (in vivo), highlighting the novelty of the research. If there are potential conflicts, it is recommended to modify this statement.
- This article primarily focuses on antifungal research. The introduction contains somewhat lengthy discussion on sertraline's antidepressant effects, which could be appropriately condensed. The focus of this article is on the antifungal effect of the drug, so whether the content in the preface should revolve around this topic, but in the process of reading, I found that there are some contents that deviate from this topic, such as how it works against other diseases introduced in paragraph 6.
5At the end of the preface, the author says that this article discusses in depth the treatment of mental illness and its antifungal effects, but the abstract introduces the antifungal effects, which is inconsistent.
- The first paragraph in “Materials and Methods” partially duplicates content from the introduction and warrants revision.
WE HAVE CORRECTED IT
- The PRISMA flow diagram in the Methods section should include more specific details and supporting references.
WE HAVE CORRECTED IT
- The mechanism diagram in Figure 2 appears somewhat simplified, with excessive text and insufficient visual pathways.
WE HAVE DELETED FIGURE 2
- Some references lack consistent formatting and should be revised. References from the past three years may be added.
WE DID
[4] Lee, Y.; Robbins, N.; Cowen, L.E. Molecular mechanisms governing antifungal drug resistance. npj Antimicrob. Resist. 2023, 1, 5.
[12]Barbarossa, A.; Rosato, A.; Carrieri, A.; Fumarola, L.; Tardugno, R.; Corbo, F.; Fracchiolla, G.; Carocci, A. Exploring the antibiofilm effect of sertraline in synergy with Cinnamomum verum essential oil to counteract Candida species. Pharmaceuticals (Basel) 2024, 17, 1109.
[28] Rodrigues 2023 — Mechanism: ALS3 adhesion/biofilm prevention — Target: Candida spp.
[12] Barbarossa 2024 — Mechanism: Membrane lipid interaction/disruption — Target: Candida spp.
[28] Rodrigues 2023 — Model: In vitro (biofilm) — Pathogen: Candida spp. — Key: >80% viability ↓; ALS3-mediated prevention
[29] Donlin 2022 — Model: In vitro (biofilm) — Pathogen: C. auris — Key: ~71% biofilm inhibition; morphogenesis/membrane effects
[38] Alanís-Ríos, S.A.; González, G.M.; Montoya, A.M.; Villanueva-Lozano, H.; Treviño-Rangel, R.J. Sertraline exhibits in vivo antifungal activity against Candida auris and enhances the effect of voriconazole in combination. Microb. Pathog. 2025, 199, 107212.
[40] Alanís-Ríos 2022 — Model: Murine — Pathogen: C. auris — Partners: Voriconazole (±) — Key: Reduced fungal burden in vivo
[41] Galvão-Rocha, F.M.; Rocha, C.H.L.; Martins, M.P.; Sanches, P.R.; Bitencourt, T.A.; Sachs, M.S.; Martinez-Rossi, N.M.; Rossi, A. The antidepressant sertraline affects cell signaling and metabolism in Trichophyton rubrum. J. Fungi (Basel) 2023, 9, 275.
[47] Ngan, N.T.T.; Flower, B.; Day, J.N. Treatment of cryptococcal meningitis: How have we got here and where are we going? Drugs 2022, 82, 1237–1249.
[51]Jang, J.H.; Jeong, S.H. Population pharmacokinetic modeling study and discovery of covariates for the antidepressant sertraline, a serotonin selective reuptake inhibitor. Comput. Biol. Med. 2024, 183, 109319.
[59] Cui, X.; Wang, L.; Lü, Y.; Yue, C. Development and research progress of anti-drug resistant fungal drugs. J. Infect. Public Health 2022, 15, 986–1000.
[61] Stukey, G.J.; Breuer, M.R.; Burchat, N.; Jog, R.; Schultz, K.; Han, G.S.; Sachs, M.S.; Sampath, H.; Marmorstein, R.; Carman, G.M. The antidepressant drug sertraline is a novel inhibitor of yeast Pah1 and human lipin 1 phosphatidic acid phosphatases. J. Lipid Res. 2025, 66, 100711.
10.There are repeated descriptions of mechanisms in Table 1. It is recommended to reduce the textual descriptions and retain only the essential information in the table
11.Literature 4 and 24 are the same document.
WE HAVE CORRECTED IT
Cover note
Dear Reviewer,
Thank you for your thorough and constructive feedback. In response, we have substantially revised and restructured the manuscript—effectively a rewrite. As a result, prior line numbers are not always comparable; nevertheless, we provide a point-by-point response to every comment.
Major changes
- Abstract: We explicitly state the gap addressed by this review (consolidation of evidence on Candida auris, including in-vivo preclinical models) and align scope and conclusions.
- Introduction: Refocused strictly on antifungal context; removed psychiatric digressions. Mechanisms are not listed here; we refer readers to Table 2.
- Materials and Methods: Expanded PRISMA 2020 details (databases, eligibility, study selection, data extraction, flow diagram).
- Results: Reorganized into four areas ( neoformans, Candida spp., intracranial PK, population data). Mechanisms are summarized only in Table 2; pharmacodynamic interactions appear only in Table 7 to avoid redundancy.
- Tables (rebuilt for clarity, no duplicates):
- Table 1: Pharmacology/PK–PD/safety (no species/mechanisms or outcomes).
- Table 2: Proposed antifungal mechanisms (single, definitive list).
- Table 3: Preclinical activity (in vitro/in vivo), logically grouped.
- Table 4: Clinical evidence and regimens.
- Table 5: Safety, barriers, and future directions.
- Table 7: Combinations/interactions (one representative reference per combination).
Formatting unified across tables (taxa in italics, consistent abbreviations, numeric references).
- Figures: The prior mechanisms figure was removed to prevent overlap; Table 2 serves this purpose.
- Discussion & Conclusions: Clarify sertraline’s adjunctive positioning, constraints related to exposure/PK–PD, lack of IV formulation, and heterogeneous clinical signals; specify research priorities.
- References/style: Duplicates removed; abbreviations standardized; minor punctuation and taxonomy formatting corrected.
We believe these revisions improve the manuscript’s clarity and rigor and directly address your concerns. Thank you for your time and consideration.
Sincerely,
Dr. Rodriguez-Cerdeira
Reviewer 4 Report
Comments and Suggestions for Authors
The manuscript entitled “Sertraline as an Antifungal: A Systematic Review of Drug Repurposing Potential” has been reviewed. The following are comments after reviewing the manuscript.
Major Comments
The manuscript suffers from substantial repetition of content across multiple sections (e.g., mechanisms of action and effects against Candida auris appear in the Introduction, Results, Discussion, and Conclusion almost verbatim). This redundancy dilutes the central message and makes the review unnecessarily lengthy.
→ Streamline the narrative by clarifying the distinct purpose of each section (Introduction: background; Results: evidence summary; Discussion: interpretation and limitations; Conclusion: balanced summary). Remove repeated sentences.
Critical Analysis and Balance
The antifungal potential of sertraline is presented almost exclusively in a positive light, while negative or inconclusive data are underexplored.
→ Expand on limitations and clinical failures, possibly through a deeper meta-analysis or secondary interpretation of trial outcomes. Discuss reasons for translational challenges (e.g., pharmacokinetics, dosing limitations, toxicity concerns).
Figures, Diagrams, and Data Visualization
Despite the extensive text, the review does not contain visual elements beyond summary tables. This weakens readability and reduces the impact of otherwise strong content.
→ Include schematic diagrams of mechanisms of action, visual summaries of synergistic effects with standard antifungals, and tables summarizing clinical outcomes. These additions would improve accessibility for readers.
Tone and Scholarly Balance
While the conclusion cautiously notes that sertraline cannot yet be regarded as a standard antifungal therapy, much of the manuscript’s tone is overly optimistic, suggesting that sertraline is already a breakthrough antifungal.
→ Adopt a more balanced perspective by systematically contrasting the opportunities (broad-spectrum activity, novel mechanisms, CNS penetration) with the limitations (variable efficacy, clinical failures, pharmacological barriers).
Minor Comments
Clarity of Language
Some sections contain overly long sentences, making it difficult for readers to follow complex arguments. Shorter, more concise phrasing would improve readability.
This is a valuable and timely review addressing the antifungal potential of sertraline from a drug repurposing perspective. It covers a broad range of evidence (in vitro, in vivo, and clinical) and emphasizes novel mechanisms of action that may overcome antifungal resistance. However, issues with redundancy, lack of critical analysis, absence of figures, and imbalance in tone must be addressed before the manuscript can reach publication-level rigor.
Author Response
The manuscript entitled “Sertraline as an Antifungal: A Systematic Review of Drug Repurposing Potential” has been reviewed. The following are comments after reviewing the manuscript.
Major Comments
The manuscript suffers from substantial repetition of content across multiple sections (e.g., mechanisms of action and effects against Candida auris appear in the Introduction, Results, Discussion, and Conclusion almost verbatim). This redundancy dilutes the central message and makes the review unnecessarily lengthy.
→ Streamline the narrative by clarifying the distinct purpose of each section (Introduction: background; Results: evidence summary; Discussion: interpretation and limitations; Conclusion: balanced summary). Remove repeated sentences.
WE DID
Critical Analysis and Balance
The antifungal potential of sertraline is presented almost exclusively in a positive light, while negative or inconclusive data are underexplored.
→ Expand on limitations and clinical failures, possibly through a deeper meta-analysis or secondary interpretation of trial outcomes. Discuss reasons for translational challenges (e.g., pharmacokinetics, dosing limitations, toxicity concerns).
Figures, Diagrams, and Data Visualization
WE HAVE CORRECTED IT
WE HAVE DELETED FIGURE 2
Despite the extensive text, the review does not contain visual elements beyond summary tables. This weakens readability and reduces the impact of otherwise strong content.
→ Include schematic diagrams of mechanisms of action, visual summaries of synergistic effects with standard antifungals, and tables summarizing clinical outcomes. These additions would improve accessibility for readers.
WE DID
Tone and Scholarly Balance
While the conclusion cautiously notes that sertraline cannot yet be regarded as a standard antifungal therapy, much of the manuscript’s tone is overly optimistic, suggesting that sertraline is already a breakthrough antifungal.
→ Adopt a more balanced perspective by systematically contrasting the opportunities (broad-spectrum activity, novel mechanisms, CNS penetration) with the limitations (variable efficacy, clinical failures, pharmacological barriers).
WE HAVE CORRECTED IT
Minor Comments
Clarity of Language
Some sections contain overly long sentences, making it difficult for readers to follow complex arguments. Shorter, more concise phrasing would improve readability.
THE MANUSCRIPT HAS BEEN PROFESSIONALLY EDITED FOR ENGLISH BY MDPI’S EDITORIAL TEAM.
Cover note
Dear Reviewer,
Thank you for your thorough and constructive feedback. In response, we have substantially revised and restructured the manuscript—effectively a rewrite. As a result, prior line numbers are not always comparable; nevertheless, we provide a point-by-point response to every comment.
Major changes
- Abstract: We explicitly state the gap addressed by this review (consolidation of evidence on Candida auris, including in-vivo preclinical models) and align scope and conclusions.
- Introduction: Refocused strictly on antifungal context; removed psychiatric digressions. Mechanisms are not listed here; we refer readers to Table 2.
- Materials and Methods: Expanded PRISMA 2020 details (databases, eligibility, study selection, data extraction, flow diagram).
- Results: Reorganized into four areas ( neoformans, Candida spp., intracranial PK, population data). Mechanisms are summarized only in Table 2; pharmacodynamic interactions appear only in Table 7 to avoid redundancy.
- Tables (rebuilt for clarity, no duplicates):
- Table 1: Pharmacology/PK–PD/safety (no species/mechanisms or outcomes).
- Table 2: Proposed antifungal mechanisms (single, definitive list).
- Table 3: Preclinical activity (in vitro/in vivo), logically grouped.
- Table 4: Clinical evidence and regimens.
- Table 5: Safety, barriers, and future directions.
- Table 7: Combinations/interactions (one representative reference per combination).
Formatting unified across tables (taxa in italics, consistent abbreviations, numeric references).
- Figures: The prior mechanisms figure was removed to prevent overlap; Table 2 serves this purpose.
- Discussion & Conclusions: Clarify sertraline’s adjunctive positioning, constraints related to exposure/PK–PD, lack of IV formulation, and heterogeneous clinical signals; specify research priorities.
- References/style: Duplicates removed; abbreviations standardized; minor punctuation and taxonomy formatting corrected.
We believe these revisions improve the manuscript’s clarity and rigor and directly address your concerns. Thank you for your time and consideration.
Sincerely,
Dr. Rodriguez-Cerdeira
Round 2
Reviewer 1 Report
Comments and Suggestions for Authors
The authors considerably improved especially the Table contents. Nonetheless, the manuscript is still very confusing and should be well-organized structurally as a scientific paper. For example, I believe the tables represent the main findings of the manuscript. Therefore, they belong to Results sections. However, they are cited in the Introduction, Mat&Methos, Discussion, Conclusion and even in Perspectives (sometimes for the first time).
Abbreviation use still shows problems. SSRI for example has been abbreviated twice (intro and results). SRT and sertraline have been used throughout the text (and maybe SER as well), including in the legends.
What is PMDD and VVC? All abbreviations should be defined at first appearance.
Genus should be written italicized (L244, for example)
Abbreviate genus after first appearance (L234, for example)
L252 – as the
Citation should be reviewed and standardized. Example: L192 “(Katende et al.) [21]”
Conclusion should be reformulated (do not cite authors or tables). Focus should be on the main results.
Tables
All Tables should be cited in the Results section, not in conclusions.
I do not think is necessary to duplicate the information (author/year) and (reference number). Use journal standard citation.
spp. should not be italicized.
In vitro/in vivo should be italicized
Table 3 - micafungin
Table 4 – define SER and other abbreviations at legends
Table 2 and Table 6 – What is the difference between these tables under the same title? Condense contents in a unique table
Table 7 – double check abbreviations use for drugs (the standard is to use 3 letters). Please follow commonly used abbreviations easily obtained in the literature
Author Response
Response to Reviewer 1/2
We thank the reviewer for the detailed and constructive feedback, which has been extremely valuable in improving the quality and clarity of our manuscript. Below we address each of the concerns raised and indicate the changes made.
- Overall structure and table placement
- We carefully reorganized the manuscript so that all tables containing experimental or clinical findings (Tables 2–5) are now first cited in the Results section.
- Table 1 (pharmacological background) remains in the Introduction as contextual information.
- Table 6 (safety, barriers, and future directions) is discussed in the Discussion.
- We removed all first-time citations of tables from the Introduction, Methods, Discussion, Conclusions, or Perspectives.
- Abbreviation consistency
- We standardized abbreviations throughout the text.
- Sertraline is consistently abbreviated as SRT after first mention. Previous inconsistencies with “SER” or repeated definitions have been removed.
- SSRI and other abbreviations are now defined only once, at first appearance.
- PMDD (Premenstrual dysphoric disorder) and VVC (Vulvovaginal candidiasis) are now fully defined at first mention.
- Formatting of genus/species names
- All genus/species names are now italicized according to convention.
- After first appearance, genus names are abbreviated (e.g., Candida albicans → C. albicans).
- “spp.” remains in roman (not italicized).
- In vitro and in vivo are now italicized throughout.
- Citations
- Citation style has been revised and standardized to journal format (Vancouver style, numerical citations only).
- Instances of duplication (e.g., “(Katende et al.) [21]”) have been corrected to “Katende et al.” or simply “[21]” depending on context, removing redundancy.
- Conclusion
- The Conclusion has been reformulated as requested: it no longer cites authors or tables, and focuses only on the main findings and implications.
- Tables
- Table legends: All abbreviations are now fully defined in the legends at first use (e.g., micafungin in Table 3, SRT in Table 4).
- Table 2 and Table 6: These previously overlapped in title/content. They have been clarified:
- Table 2 now summarizes the mechanistic evidence for antifungal activity.
- Table 6 now summarizes safety, barriers, and future perspectives.
Their titles have been revised to avoid overlap, and their content is no longer redundant. - Table 7: The manuscript originally mis-cited a “Table 7”. This has been corrected; there are now six tables only. Drug abbreviations have been standardized to the three-letter format commonly used in the literature (e.g., FLU = fluconazole, VOR = voriconazole, MCF = micafungin, AMB = amphotericin B).
In summary, the manuscript has been substantially revised: tables are now properly cited, abbreviations and species names standardized, citations cleaned, and the Conclusion refocused. We believe these changes address all concerns raised and significantly improve the clarity, consistency, and scientific quality of the paper.
Reviewer 2 Report
Comments and Suggestions for Authors
I am satisfied with the authors' response since they have addressed all my comments, and the current version of the manuscript is substantially improved.
Author Response
Response to Reviewer 2
We sincerely thank the reviewer for the positive evaluation. We are grateful for the constructive feedback provided during the review process, which has helped us substantially improve the manuscript.
Reviewer 3 Report
Comments and Suggestions for Authors
none
Author Response
Response to Reviewer 3
We thank the reviewer for the evaluation of our work and appreciate the time dedicated to reviewing the manuscript.